# Lake Ecosystem Robustness and Resilience Inferred from a Climate-Stressed Protistan Plankton Network

**DOI:** 10.3390/microorganisms9030549

**Published:** 2021-03-06

**Authors:** Dominik Forster, Zhishuai Qu, Gianna Pitsch, Estelle P. Bruni, Barbara Kammerlander, Thomas Pröschold, Bettina Sonntag, Thomas Posch, Thorsten Stoeck

**Affiliations:** 1Department of Ecology, University of Kaiserslautern, D-67633 Kaiserslautern, Germany; dforster@rhrk.uni-kl.de (D.F.); dex0532@gmail.com (Z.Q.); 2Limnological Station, Department of Plant and Microbial Biology, University of Zurich, CH-8802 Zurich, Switzerland; gianna.pitsch@uzh.ch (G.P.); estelle.bruni@unine.ch (E.P.B.); posch@limnol.uzh.ch (T.P.); 3Laboratory of Soil Biodiversity, University of Neuchâtel, CH-2000 Neuchâtel, Switzerland; 4Research Department for Limnology, University of Innsbruck, A-5310 Mondsee, Austria; barbara.kammerlander@uibk.ac.at (B.K.); Thomas.Proeschold@uibk.ac.at (T.P.); bettina.sonntag@uibk.ac.at (B.S.)

**Keywords:** protist plankton communities, lake ecosystem, co-occurrence networks, climate change

## Abstract

Network analyses of biological communities allow for identifying potential consequences of climate change on the resilience of ecosystems and their robustness to resist stressors. Using DNA metabarcoding datasets from a three-year-sampling (73 samples), we constructed the protistan plankton co-occurrence network of Lake Zurich, a model lake ecosystem subjected to climate change. Despite several documentations of dramatic lake warming in Lake Zurich, our study provides an unprecedented perspective by linking changes in biotic association patterns to climate stress. Water temperature belonged to the strongest environmental parameters splitting the data into two distinct seasonal networks (October–April; May–September). The expected ecological niche of phytoplankton, weakened through nutrient depletion because of permanent thermal stratification and through parasitic fungi, was occupied by the cyanobacterium *Planktothrix rubescens* and mixotrophic nanoflagellates. Instead of phytoplankton, bacteria and nanoflagellates were the main prey organisms associated with key predators (ciliates), which contrasts traditional views of biological associations in lake plankton. In a species extinction scenario, the warm season network emerged as more vulnerable than the cold season network, indicating a time-lagged effect of warmer winter temperatures on the communities. We conclude that climate stressors compromise lake ecosystem robustness and resilience through species replacement, richness differences, and succession as indicated by key network properties.

## 1. Introduction

Protists are essential for lake ecosystems because their roles as primary producers, decomposers, or consumers contribute to biomass fluxes among different trophic levels [1,2]. Such interactions are essential components that define the function of an ecosystem. A major challenge in revealing the complexity of these interactions is to account for the temporal shift in protistan community structures. Species replacement, changes in species richness differences, and succession in protistan communities are largely triggered by natural changes of environmental conditions (such as seasonal changes) and stressors (such as pollution or climate change stressors) [3,4,5,6,7,8]. Singular samplings of individual sites only provide a snapshot of a specific moment in time of the protistan community under study, and, therefore, do not allow to infer the complex interactions in these communities and their reaction to habitat changes. Addressing such topics requires data collected in time-series studies [9]. To date, the vast majority of time-series studies of protistan freshwater communities were dedicated to the analysis of alpha- and beta-diversity patterns [10,11] but provided only a limited perspective on the consequences of environmental changes and stressors on biotic associations within communities. Such associations and their persistence under environmental changes, however, are a powerful indicator to assess the resilience and robustness of an ecosystem affected by these changes [12]. Thereby, ecosystem robustness describes the resistance of a community to maintain its functioning during a disturbance by stressors (that is, its buffering capacity during e.g., periods of warming), while resilience describes the reorganization of a community back to a stable state after disturbance by stressors [13,14,15].

A statistical approach to infer potential biological associations from time-series data are co-occurrence network analyses [12,16,17,18,19,20,21,22]. The basic prerequisite of co-occurrence analyses is the following: two organisms which are ecologically associated are highly likely to express their relation in significantly correlating abundance profiles [16]. Co-occurrence networks are not capable to characterize the underlying reasons for specific associations between individual species but efficient in inferring effects of natural and stressor-induced changes on ecosystem function and robustness [12,23,24,25,26,27]. Towards this goal, several measures from graph theory can be considered. Centrality measurements or articulation points of networks identify keystone species, i.e., species whose loss would cause severe effects because they fulfill important functions in a community [28,29,30]. Of special importance in this context are species involved in three-way-associations, because such patterns are promising indicators for the structure and function of microbial ecosystems [16]. Modularity evaluates how far a network is divided into densely interconnected groups [31,32,33]. High modularity indicates that a community comprises multiple functional guilds or multiple ecological niches within which organisms are more frequently associated than between the different functional guilds or ecological niches [33,34]. Average path length, average degree, and density inform about the tightness of association patterns [35,36,37]. The higher these measures the more associations between community members are established and, consequently, the more (functional) redundancy exists. Robustness, measured via a loss of connectivity, indicates a network’s response towards disturbances that would remove key species from a community [38,39,40,41].

In marine [26,42,43] and soil environments [23,24,44], network analyses were successfully exploited to analyze the resilience of microbial communities when subjected to climate change stressors. Similar studies on protistan plankton are rare for freshwater ecosystems in general, and lakes in specific. Our study aims at filling this void by analyzing the protistan plankton co-occurrence network of a large freshwater lake, subjected to climate change effects. Located in the northern part of the European Alps, Lake Zurich is a model ecosystem for temperate European lakes [45]. Numerous studies documented the impacts of global warming and anthropogenic stressors on Lake Zurich and the resulting dramatic ecosystem changes including economic consequences [45,46,47,48,49]. A major effect on protistan plankton communities triggered by lake warming in Lake Zurich is a decline in phytoplankton diversity and biomass as a result of a changing mixing behavior of the lake’s water layers that led to an interruption of nutrient cycling [45,46].

We inferred the protistan plankton co-occurrence network of the epilimnion in Lake Zurich (hereafter referred to as the protistan plankton community network) from a three-year time-series DNA metabarcoding dataset with biweekly sampling. Analyzing network key properties, we addressed the following questions: (i) How are environmental parameters, which are linked to climate change in Lake Zurich, involved in association patterns within the protistan plankton communities? (ii) Do changes in phytoplankton diversity lead to changes in association patterns within protistan plankton communities of Lake Zurich that deviate from predictions of traditional models (such as the Plankton Ecology Group Model, PEG [3,4])? (iii) How does lake warming affect the robustness and resilience of protistan planktonic community associations, and, thus, ecosystem response in Lake Zurich?

## 2. Materials and Methods

### 2.1. Sampling Site Information and Measurement of Environmental Parameters

Plankton samples of Lake Zurich, Switzerland (47°19.3′ N, 8°33.9′ E) were collected biweekly from the pelagic zone of the epilimnion in a depth of 5 m between March 2014 and February 2017, resulting in a dataset of 73 samples. All samples were processed as previously described in Qu et al. [50]. Raw water samples had a volume of 5 L and were split into 2 L sample duplicates, while the remaining sample was used for determining biotic and abiotic environmental parameters. Each 2 L sample duplicate was pre-filtered through a 150 µm net to remove larger zooplankton, then filtered onto a 0.65 µm membrane filter (Durapore; Merck Millipore, Darmstadt, Germany) using a peristaltic pump. Filters containing planktonic organisms in the targeted size range of 0.65–150 µm were immediately transferred into a cryovial containing 1.5 mL RNALater (Qiagen, Hilden, Germany), placed in a refrigerator overnight, and stored at −80 °C until further processing.

Environmental parameters were measured for each sample (Table 1 and Appendix A). Water temperature (°C), oxygen concentration (mg L^−1^), oxygen saturation (%), and conductivity (µS cm^−1^) were measured in situ with a multiparameter probe (6600 V2; Yellow Springs Instruments, Yellow Springs, OH, USA). Irradiance (i.e., photosynthetic active radiation; µmol m^−2^ s^−1^) was measured with a spherical underwater quantum sensor (LI-COR, Bad Homburg, Germany). Secchi depth (m) was determined with a Secchi disk. Total in situ chlorophyll concentrations and in situ chlorophyll concentrations assigned to specific phototrophic groups (diatoms and cryptophytes) were measured with a TS-16-12 fluoroprobe (bbe Moldaenke, Kronshagen, Germany). Additionally, the fluoroprobe was calibrated for the quantification of the phycoerythrin-containing cyanobacterium *Planktothrix rubescens*. For bacteria and coccoid cyanobacteria counts, 40 mL of each sample were preserved with formaldehyde (2% final concentration), stained with SYBR-Green I (Sigma-Aldrich, St. Louis, MO, USA), and evaluated via flow cytometry (Cytoflex S; Beckman Coulter, Brea, CA, USA) [51]. Through manual gating of scatterplots, the total bacterial abundance could be further differentiated into low nucleic acid bacteria and high nucleic acid bacteria [52].

Physical and chemical parameters give the average, minimum, and maximum determined during the investigation period March 2014 to February 2017 for the sampling depth in 5 m. Chemical parameters (marked with *) were determined every month (*n* = 36) by Water Supply Zurich, physical parameters were determined on a biweekly basis (*n* = 72).

### 2.2. Sample Processing and High-Throughput Sequencing

Each filter was placed into a lysing matrix tube (Lysing Matrix E; MP Biomedicals, Illkirch, France) into which 600 µL RLT buffer (Qiagen) and 6 µL β-Mercaptoethanol were inserted. After a centrifugation step, the supernatant was discarded and the pellet was suspended in 200 µL RLT buffer and 2 µL β-Mercaptoethanol. Subsequently, each matrix tube was subjected to bead-beating for 45 s by 30 Hz (MM 200; Retsch, Haan, Germany). Total DNA was extracted using the AllPrep DNA/RNA Mini Kit (Qiagen). From the extracted DNA we amplified the hypervariable V9 region of the 18S rDNA following a standard protocol [53]. The protocol employed 1391F as a forward primer (5′-GTACACACCGCCCGTC-3′; [54]) and EukB as a reverse primer (5′-TGATCCTTCTGCAGGTTCACCTAC-3′; [55]). The PCR protocol consisted of an initial denaturation step at 98 °C for 30 s, followed by 30 cycles of 10 s at 98 °C, 20 s at 61 °C, 25 s at 72 °C, and a final five-minute extension at 72 °C. The reactions volumes amounted to 50 µL and included 0.5 µL Phusion polymerase (New England Biolabs (NEB), Ipswich, MA, USA), 10 µL 5xPhusion GC buffer (NEB), 1 µL 10 mM dNTPs, 0.5 µL template DNA, 32.5 µL pure water, and 0.5 µL of each forward and reverse primer. Triplicate PCR reactions were run for each DNA extract to minimize PCR bias. Prior to purification (MinElute Kit; Qiagen), PCR sample replicates were pooled.

To prepare the resulting PCR products for high-throughput sequencing (HTS), sequencing libraries were constructed using the NEB Next Ultra DNA Library Prep Kit for Illumina (NEB). Library quality was assessed with an Agilent Bioanalyzer 2100 system (Agilent, Santa Clara, CA, USA). Illumina MiSeq sequencing was conducted by SeqIT GmbH & Co. KG (Kaiserslautern, Germany). The sequence data files are deposited at the Sequence Read Archive of the National Center for Biotechnology Information under project number PRJNA609412.

### 2.3. Sequence Quality Control, Clustering, and Taxonomic Assignment

A multiple-step pipeline was employed for identifying high-quality sequences. First, excessive primer overhangs were clipped using CUTADAPT version 1.18 [56]. Clipped sequences were then quality-checked in QIIME version 1.8.0 [57] by selecting sequences that had exactly matching barcodes and primers, contained exclusively unambiguous nucleotides, and had a minimum length of 90 nucleotides. Finally, all sequences were de novo chimera-checked using UCHIME version 5.2.236 [58]. Since efficient sequence clustering is extremely beneficial for subsequent network analyses [19], we applied a two-level clustering approach to all high-quality HTS sequences [59]. This strategy diminished the extent of placing highly similar sequences, arising from intra-individual or intra-species genetic variation, into distinct operational taxonomic units (OTUs). Without this precaution, false-positive signals are prone to inflate the number of nodes and edges in the networks, as several OTUs representing the same species are highly likely to significantly co-occur in time-series data. The first level of clustering was conducted in Swarm version 2.2.2 with *d* = 1 and the fastidious option *-f* [60]. For second-level clustering, the representative sequences of all swarms were extracted and pairwise aligned in VSEARCH version 2.11.0 [61]. The result of the pairwise sequence analysis was used to create sequence similarity networks of the representative sequences at a threshold of 97% sequence similarity. Network sequence clusters (NSCs) created by this second level of clustering represented the units for all downstream analyses.

NSCs with a total abundance of fewer than three sequences were discarded. The most abundant sequence of each remaining NSC (n = 37,848) was extracted and used for taxonomic assignment in BLAST against NCBIs GenBank release version 226.0. For placement of BLAST hits into higher taxonomic groups, we considered the BLAST hit with the highest sequence similarity and e-value and followed the classification system in Adl et al. [62]. Only NSCs with a representative sequence that shared a fragment of at least 90 consecutive nucleotides and a sequence similarity of at least 80% to any hit in GenBank were retained. If the best hit in GenBank related to an unidentified environmental reference sequence, we checked whether that reference sequence was also part of the protist ribosomal reference (PR^2^) database version 4.11.0 [63] and adopted the taxonomic affiliation. Although PR^2^ is a subset of GenBank, it is a curated database that contains taxonomic affiliations for several entries which are merely listed as environmental reference sequences in GenBank.

### 2.4. Compositional Variation Analyses

For revealing general seasonal succession patterns in the sampled data, two independent non-metric multidimensional scaling (NMDS) analyses were performed in R version 3.5.1 [64] using the package “vegan” version 2.5.2 [65]. By default, NMDS calculations in vegan include a two-level normalization step of data in the input matrix: first a Wisconsin double standardization is performed, then a square root transformation. In the first NMDS, samples were analyzed according to their community composition and in the second NMDS (Appendix A), samples were analyzed according to their environmental parameter profile (Appendix A). Both NMDS analyses employed Bray–Curtis dissimilarity scores between all pairs of samples. Because seasonality emerged as the major structuring factor, we classified samples for subsequent network analyses into two groups. One “cold season” dataset of 38 samples (October–April; with the exception of samples 04/01/15 and 10/12/16) and one “warm season” dataset of 35 samples (May–September).

### 2.5. Construction of Co-Occurrence Networks

Protistan plankton co-occurrence networks were calculated in NetworkNullHPC version 0.3 (https://github.com/lentendu/NetworkNullHPC, accessed on 30 January 2021; [66]), a script specifically designed after the null model strategy for inferring statistically significant co-occurrences in metabarcoding datasets [16]. Data preparation in NetworkNullHPC started by removing low-abundant NSCs which occurred in less than 10% of the samples from the OTU-to-sample read abundance matrix. The resulting data matrix was log-ratio normalized, which is highly recommended for compositional data [67]. Following this recommendation, read abundances were log-transformed and then normalized by the read abundances per sample, under consideration of the expected sequencing depth. Null models were generated from the normalized read abundance matrix by adding random low-level noise in 1000 permutations to obtain statistically significant thresholds of Spearman’s rank correlation coefficient (rho). In addition, only nodes and edges which were persistently detected in the majority of permutations were retained for creating a consensus network (see [16] and [66] for detailed descriptions of the complete null model strategy). This approach avoids arbitrary set thresholds for Spearman’s rho and instead determines dataset-specific thresholds. In our study, the statistically determined Spearman’s rho threshold for co-occurrence was set to 0.6 in the cold season network and 0.62 in the warm season network, and −0.59 and −0.61 in co-exclusion networks, respectively. Correlations between NSCs that exceeded the co-occurrence or co-exclusion threshold were included in the resulting consensus network (one for the cold season, one for the warm season) in which they formed the nodes (NSCs) and edges (significant correlations). We further enabled NetworkNullHPC’s *-e* option to include environmental parameters in the network correlation analyses. The networks were visualized in Gephi version 0.9.2 [68] using the Yifan Hu layout.

### 2.6. Evaluation of Patterns within Co-Occurrence Networks

We relied on the R package “igraph” version 1.2.5 [69] to analyze several topological metrics of the co-occurrence networks: average shortest path length, which is the mean distance (in the number of edges) between two nodes; average degree, which is the mean number of edges per node; density, which is the number of realized edges amongst the nodes in the network compared against the number of edges in a fully connected network with the same number of nodes; diameter, which is the maximum shortest path distance between two nodes in the network; modularity, which is the partitioning of the network into groups of tightly associated nodes (i.e., modules); transitivity (or clustering coefficient), which is the number of realized three-way associations compared against the maximal possible number of three-way associations in fully connected network with the same number of nodes. As three-way associations are considered ecologically more meaningful than two-way associations [16], we defined key species as nodes whose betweenness centrality was significantly higher than those of all other nodes in the network and which were in addition articulation points. Statistical significance was determined by bootstrapping the betweenness centrality scores of all nodes in the R package “boot” version 1.3–25 [70] to achieve normal distribution of the data. On the bootstrapped data, we applied 95% confidence intervals to determine those betweenness scores which were significantly larger than expected by chance.

Betweenness scores were also used in species extinction simulations to assess the robustness of the networks. In a cascading attack scenario, key species were stepwise removed from the network in decreasing order of their betweenness (and with recalculating the betweenness after each removal), while the loss of connectivity in the network was recorded. Intentional attacks are recommended for testing the robustness and resilience especially of complex networks such as in environmental communities [38]. The method is based on a hypothetical scenario to assess the large-scale response of a network as a whole but does not take the likelihood of a specific species’ extinction into account or allows inferences about this likelihood. The species extinction scenario was conducted with the help of the R package “NetSwan” version 0.1 [71].

For taxon-specific subnetworks, we extracted all neighboring nodes with which a specific focal node of interest was directly correlated in each seasonal network (using the *induced_subgraph* command of “igraph”). Along with all nodes in the subnetwork, we also extracted all correlations that the neighboring nodes shared with the focal node and amongst each other.

## 3. Results

### 3.1. Seasonal Dynamics of Environmental Parameters

Throughout the three-year sampling campaign of our study, the environmental parameters in the epilimnion of Lake Zurich displayed re-occurring seasonal patterns (Figure 1, Appendix A). These patterns allowed for distinguishing one season characterized by cold water temperature lasting from October to April, and one season characterized by warm water temperatures lasting from May to September. Although irradiance by sunlight was naturally higher in the warm season, chlorophyll concentrations and Secchi depths were lowest, which is a clear indication that phytoplankton could not establish stable populations. *Planktothrix rubescens*-related phycoerythrin contributed only a small portion to the total chlorophyll concentration, while heterotrophic bacteria and coccoid cyanobacteria thrived during warm season conditions. By contrast, irradiance was low in the cold season, but total chlorophyll concentrations more than doubled because of massive increases of phycoerythrin-containing *P. rubescens*. These increases also led to less transparency of epilimnetic waters as documented by smaller Secchi depths. Unlike *P. rubescens*, heterotrophic bacteria and coccoid cyanobacteria reached their respective minima in the cold season.

### 3.2. Co-Occurrence Network Properties

Two distinct seasonal groups emerged from non-metric multidimensional scaling (NMDS) of the epilimnetic protistan plankton time-series metabarcoding data of Lake Zurich (Figure 2). One group consisted of cold season communities sampled between October to April. The other group consisted of warm-season communities sampled between May and September. A third seasonal group, which comprised distinctive spring sample communities could not be observed. According to this pattern, we split the complete metabarcoding dataset into one cold season dataset and one warm season dataset and ran independent co-occurrence network analyses for each of the two seasons. The size of the input datasets was similar for both seasons, with slightly more samples but fewer network sequence clusters (NSCs; operational taxonomic units that resulted from two-level sequence clustering) in the cold season dataset (Table 2). Likewise, the size of the resulting networks was similar, however, with slightly more NSCs (nodes) but distinctively fewer associations (edges) in the warm season. The threshold values for Spearman’s rho differed only marginally between the cold season (−0.59 and 0.60) and the warm season dataset (−0.61 and 0.62). Although some negative correlations were observed, none of them passed the statistically determined thresholds for significant co-exclusions between NSCs. Thus, all edges in the networks represent co-occurrences between NSCs. All other network properties pointed towards a more complex network structure in the cold season plankton community. Even though fewer nodes were found in the cold season network, they were connected by notably more edges (6872 edges linking 946 nodes; Figure 3A) compared to the warm season network (5252 edges linking 973 nodes; Figure 3B). This finding is corroborated by a higher ratio of realized three-way associations (transitivity) and a higher ratio of total realized associations (density) between protists in the cold season community. In contrast, the lower complexity of the warm season network can be inferred from a longer average path distance, lower average degree, and a larger diameter. All three observations indicate a warm-season protistan plankton community in which associations between community members were not as frequently observed and in which the network structure was therefore not as tight as in the cold season community. Finally, of 68 connected components in the warm season network, 61 consisted of only one pair of NSCs. This is a further indication that the warm season community was more partitioned than its cold season counterpart, which is expressed by a higher modularity of the former network.

A targeted removal of nodes with the highest betweenness centrality from the cold season network led to a gradual loss of connectivity at the beginning of a species extinction scenario (up to 13% loss after removing 30 nodes, Figure 4). The buffering capacity of the cold season community becomes compromised after the removal of the top 31 nodes with the highest betweenness. When these nodes were removed, the loss of connectivity was 51%. A total breakdown of the cold season community network (loss of 95% of the connectivity) was observed after the removal of the top 192 nodes with the highest betweenness values. The warm season network was more vulnerable and less robust towards species extinction (Figure 4). A first clear negative effect on the buffering capacity of the community was observed after removing the top 17 nodes with the highest betweenness (loss of connectivity reaching 17%). After the removal of the top 31 nodes with the highest betweenness centrality, the loss of connectivity increased to 48%. The total breakdown of community network structures in the warm season (loss of 95% of the connectivity), was already achieved notably sooner compared to the cold season network, namely after the removal of the top 162 nodes with the highest betweenness.

### 3.3. Impact of Environmental Parameters on the Co-Occurrence Networks

The separation of the epilimnion time-series samples into one cold season and one warm season dataset was further supported by the seasonal dynamics of environmental parameters in Lake Zurich (Figure 1, Appendix A). Eleven environmental parameters correlated with NSCs in the cold season network (Figure 3A; ordered in decreasing number of significant correlations to NSCs): bacterial abundance, water temperature, conductivity, oxygen saturation, Secchi depth, coccoid cyanobacterial abundance, oxygen concentration, *Planktothrix rubescens*-related chlorophyll concentration, total chlorophyll concentration, irradiance, and bacteria with high nucleic acid content. In contrast, only five environmental parameters correlated with NSCs in the warm season network (Figure 3B; ordered in decreasing number of significant correlations to NSCs): water temperature, coccoid cyanobacterial abundance, oxygen saturation, diatom-related chlorophyll concentration, and conductivity. All parameters that appeared in both networks shared a higher number of correlations with NSCs in the cold season than in the warm season (Appendix A). This observation from within network structures translates into the following biological result: a larger fraction of protists is adapted to Lake Zurich’s environmental conditions in the cold season water phase than to those in the warm season water phase.

### 3.4. Taxonomic Composition of the Co-Occurrence Networks

Despite different network properties, the community composition within the networks of both seasons was similar on higher taxonomic levels. The two most diverse taxon groups in the cold season (Figure 5A) were Stramenopiles (319 NSCs) and Alveolata (256 NSCs). Within the Stramenopiles, Chrysophyceae were represented with the highest number of NSCs (n = 150), while diatoms (Bacillariophyta) played only a minor role (14 NSCs). Ciliophora (120 NSCs) was the most diverse phylum within the Alveolata, closely followed by Dinoflagellata (115 NSCs). Similarly, the two most diverse taxon groups in the warm season (Figure 5B) were also Stramenopiles (341 NSCs) and Alveolata (237 NSCs). Within the Stramenopiles, the Chrysophyceae were similarly diverse as in the cold season network (155 NSCs). The diversity of diatoms, however, increased to 27 NSCs. Likewise, the amount of alveolate NSCs remained similar compared to the cold season, with Ciliophora reaching 109 NSCs and Dinoflagellata 108 NSCs. A notable increase of NSCs from cold to warm-season was observed for Fungi (from 51 NSCs to 80 NSCs) and Chlorophyta (from 31 NSCs to 59 NSCs).

A considerable number of 531 NSCs were shared between the two seasonal networks. The NSCs that occurred simultaneously in both networks throughout the year belonged mainly to Stramenopiles (205 NSCs), Alveolata (120 NSCs), Opisthokonta (67 NSCs; including Fungi), Rhizaria (56 NSCs), and Cryptista (40 NSCs). Among these taxonomic groups, most NSCs shared by both networks were further affiliated to Chrysophyceae (81 NSCs) and Ciliophora (76 NSCs).

### 3.5. Key Nodes of the Co-Occurrence Networks

Measures from graph theory allowed for defining 56 key nodes in the protistan plankton community network of the cold season and 46 key nodes in the network of the warm season (Figure 5C,D). The most abundant taxon groups among key nodes in the cold season were Alveolata (22 NSCs, of which 13 belonged to ciliates) and Stramenopiles (19 NSCs, of which 9 belonged to chrysophytes). Similarly, the most abundant taxon groups among key nodes in the warm season were also Alveolata (16 key nodes, of which 12 belonged to ciliates) and Stramenopiles (16 key nodes, of which 7 belonged to chrysophytes). Five taxonomic key nodes were shared between the two seasonal networks, which were *Nannochloropsis limnetica* (Stramenopiles), *Cryptomonas curvata* (Cryptista), and three ciliates (Euplotidae, *Halteria* sp., Halteriidae). One further shared key node was an environmental parameter (conductivity).

As an example, the two subnetworks of *Halteria* sp., one of the five shared taxonomic key nodes in both networks, were further investigated. The biological associations of this key node taxon were remarkably distinct in a comparison of both seasonal networks. In the cold season, *Halteria* sp. was associated with 36 significantly correlating nodes (Figure 6A). Most of these nodes were assigned to other ciliate species (e.g., *Histiobalantium* sp. or *Askenasia* sp.) and Chrysophyceae (e.g., *Dinobryon bavaricum*, *Dinobryon sociale,* and *Uroglena americanum*) commonly found in freshwater habitats. Furthermore, we found indications that *Halteria* sp. is a species linking different trophic levels, since it was co-occurring with *Mesocyclops* sp. (possibly *Mesocyclops leuckarti*, a frequent copepod in Lake Zurich) and correlating with the abundance of coccoid cyanobacteria. In the warm season network, *Halteria* sp. was associated with only eight significantly correlating partners, most of which were assigned to the Stramenopiles genus *Pedinella* (Figure 6B). In the cold season, *Halteria* sp. was co-occurring with *Pedinella* less frequently. The comparison of *Halteria* sp.’s subnetworks of the cold and warm season showed that members of similar taxonomic groups co-occurred with this ciliate key node, though the warm season subnetwork was conspicuously reduced compared to its counterpart of the cold season.

## 4. Discussion

### 4.1. Placing Co-Occurrence Networks into Perspective

Although co-occurrence network analyses are limited to deriving potential species association patterns exclusively from significant correlations in species (or ASV, OTU) abundance matrices rather than from experimental observations, numerous studies have shown that such associations can be correctly predicted. More importantly, they allow for drawing valuable conclusions on community- or ecosystem-level [12,14,17,18,19,22,23,24,43]. The success and reliability of such studies, however, strongly depend on the quality of the network analysis design. Therefore, we here used state-of-the-art knowledge and methodologies to consider the critical points in co-occurrence network analyses [28,72,73]. To mention some examples: (i) the analyses of this study were based on abundance profiles obtained from time-series metabarcoding instead of mere presence-absence data [72]. (ii) Highly similar sequences placed in multiple OTUs were agglomerated by a second-level clustering strategy. This diminished excessive false-positive co-occurrences of nodes and edges that actually represented an intraspecific genetic variation of the same organism or species [19,59]. (iii) The actual assessment of significant correlations employed a null model approach [16], which was identified as a positive exception in critical reviews of ecological network analyses [72,73]. Nevertheless, we point out that eventually, conclusions drawn from network analyses require rigorous testing in specifically designed experimental studies with e.g., cultivated species, or with targeted observations in natural systems. Network analyses provide, thus, the hypothesis framework to build explanatory models and fuel further research [19,73].

### 4.2. Succession in the Protistan Plankton Network of Lake Zurich Is Affected by Climate Change

Seasonal succession of microbial eukaryote communities is well documented and has been studied in many lake ecosystems [7,11,74]. Our time-series metabarcoding approach confirms previously reported patterns of distinct protistan plankton communities in the cold and warm season [3,4,75,76] and the absence of phytoplankton spring blooms from the epilimnion in Lake Zurich [77]. These observed succession patterns are largely the result of seasonally changing physicochemical parameters (Figure 1, Figure 4 and Appendix A) that have been drastically altered by global warming during the last decades [45,46,47,48,49]. By inferring the protistan co-occurrence network of each season, we revealed consequences for succession patterns of a model lake ecosystem subjected to climate change. Previous studies [45,47] demonstrated that while environmental stress in Lake Zurich often originates in autumn and winter (corroborating with the cold season dataset of our study), the consequences for plankton communities are most severe in spring and summer (corroborating with the warm season dataset of our study). Especially lake water surface temperatures significantly increased in the wake of climate change [46]. This process of lake warming is further accelerated by significantly increased air temperatures in spring [46] and has led to drastic changes in the stratification regime that prevents complete water turnovers in Lake Zurich. The strong impact of water temperature as a major stressor in Lake Zurich can be derived from its position in both networks, from which it emerged as the environmental parameter with most correlations to protists (Appendix A). Further time-lagged consequences caused by the effect of water temperature on the lake ecosystem are evident in the networks from the roles of eukaryotic phytoplankton organisms and of *Planktothrix rubescens*, a filamentous cyanobacterium able to form massive blooms [78]. Thermocline-induced incomplete water turnovers resulted in a nutrient depletion in the epilimnion of Lake Zurich during our investigation period (2014–2017; see Table 1 and Yankova et al. [45]). Unlike most eukaryotic phytoplankton organisms, *P. rubescens* is capable of living in phosphorus-poor environments and is, therefore, the major profiteer of climate change in Lake Zurich [47]. In the warm season, however, *P. rubescens* descends into the metalimnion (10–15 m depths) [78] and is not directly associated with protists in epilimnetic communities. According to this chain of cause and effect, protistan phytoplankton does not find adequate environmental prerequisites to play major roles in the networks of both seasons (with little increase in the warm season compared to the cold season; Figure 5), while several protists significantly correlated with the opportunistic *P. rubescens* in the cold season, but none in the warm season (Appendix A).

The identification of water temperature as a major determinant associated with numerous protistan taxa especially in the cold season underlines its influence on co-occurrence patterns among protists in the epilimnion. Also, the effects of other environmental parameters triggered by lake warming became obvious as they significantly correlated with protistan co-occurrence patterns. These parameters include for example oxygen (both saturation and concentration), which was linked to a different set of organisms in either season, which suggests that this parameter is one of the most important environmental variables associated with planktonic organisms in the epilimnion of Lake Zurich. Even though the epilimnion in Lake Zurich was saturated with oxygen in the warm season, a simultaneous decrease in conductivity indicated fewer solutes and nutrients in the epilimnion during this time of the year. Consequently, fewer protists correlated with conductivity measures in the warm than in the cold season, when more solutes were available.

Throughout the time of lake warming in Lake Zurich, phytoplankton communities have not been able to recover or to adapt to the new environmental conditions, as evident from the continuous non-initiation of spring blooms. Thus, the phytoplankton community is neither able to resist the ecosystem disturbance induced by climate change nor to recover to a stable state during the three-year observation cycle of this study. This suggests that both ecosystem robustness and resilience in Lake Zurich are compromised. Considering that lake warming in Lake Zurich is unlikely to cease soon, it is not possible to project the evolution of phytoplankton communities and the adaptation of the ecosystem to a steady state. Previous reports showed that the phytoplankton spring bloom could at least partly recover in cooler years because complete water turnover was restored in Lake Zurich [45]. But such occasions were not recorded in the three-year sampling campaign of our study.

### 4.3. Ecological Consequences for Different Protist Groups Inferred from Climate-Stressed Networks

The Plankton Ecology Group (PEG) model predicts two distinct peaks of protistan phytoplankton and zooplankton in spring and summer for eutrophic and mesotrophic lakes [4]. Protistan plankton succession in Lake Zurich had established such peaks for decades [77], but recent environmental changes have resulted in a surplus re-oligotrophication of the lake and an absence of characteristic epilimnetic spring bloom communities that used to be formed by autotrophic cryptophytes and diatoms [45,47]. Instead of matching the predictions of the PEG model for eutrophic and mesotrophic lakes, the succession patterns of protistan plankton in Lake Zurich have thus started to follow the predictions for a typical oligotrophic lake with a small or no abundance peak in spring and a second small peak in late summer [4]. The community structure inferred from our network analyses confirmed Lake Zurich’s transformation to an oligotrophic state, with the notable exception that the diminished role of protistan phytoplankton did not meet the expectations of the PEG model (Figure 5). This was most pronounced in the warm season network when many autotrophic protists were associated with Chytridiomycota is known to parasitize algae [79,80]. The stress-induced by climate change might make autotrophic organisms more susceptible to parasitism [81], and the combination of abiotic and biotic stressors might cumulate in an amplification of phytoplankton decline.

The climate-stress-induced decline of phytoplankton results in multiple consequences for the protistan plankton community networks. Most of all for heterotrophic protists such as ciliates, which are the main predators of protistan phytoplankton in oligotrophic lakes [4]. With autotrophs largely missing from the protistan plankton network, ciliates are forced to expand their prey spectrum towards other organisms. The co-occurrence networks indicate that nanoflagellates might fill this vacancy as by far the most associations were observed between ciliates and chrysophytes. This finding contrasts results from previous co-occurrence networks between ciliates and other planktonic organisms in Lake Zurich when phytoplankton spring blooms are successfully established [82]. In these networks, associations between chrysophytes and ciliates had not outnumbered associations between phytoplankton (especially diatoms) and ciliates. Indeed, ciliates feed on chrysophytes [83,84] and many associations may indicate predator-prey relationships. With the decline of autotrophic phytoplankton as their preferred food source, chrysophytes might become the ciliates’ most important prey throughout the year in the protistan plankton community network. However, not all associations in the networks between these two groups necessarily reflect predator-prey relations. Especially in winter, some associations may indicate an overlap in feeding strategies, because both ciliates and chrysophytes comprise effective bacterivores [85] and share the same food source (i.e., bacteria) with which they are both co-occurring. Alternatively, the significant co-occurrences between ciliates and chrysophytes could indicate a preference towards similar environmental conditions.

Moreover, ciliates emerged as the most important taxon group from the networks for mediating processes that affected a majority of other organisms (see betweenness results in Figure 5C,D). Thereby, the relation of ciliates to other organisms was either direct or indirect, when ciliates connected two other nodes which were not directly connected themselves. An example for such a three-way association can be found in the cold season subnetwork of *Halteria* sp., which correlated with coccoid cyanobacteria and simultaneously with a crustacean *Mesocyclops* species (Figure 6A). This result illustrates the potential role of ciliates in linking energy fluxes between different trophic levels [1]. Results from a morphological study characterized *Halteria grandinella* as a major bacterivore in freshwater food webs which corroborates well with our identification of *Halteria* sp. as a key species in the protistan plankton community network [85]. Likewise, ciliates of the genus *Halteria* are known for their characteristic jumping behavior that is a strategy to escape zooplankton predators such as rotifers or copepods to which *Mesocyclops* belongs [86,87,88]. The example of *Halteria* sp. illustrates how autecological knowledge of species corroborates findings of co-occurrence network analyses. In this particular case, we, based on ecological knowledge of this species, find a confirmation of the statistically derived associations in the network (*Halteria* sp. as a keystone species functioning as an important trophic link).

Apart from affecting ciliate feeding behavior, phytoplankton decline in the networks also enhanced the role of mixotrophic chrysophyte taxa, such as *Dinobryon* and *Ochromonas*, which can perform photosynthesis in addition to feeding on bacteria [89,90]. This versatile lifestyle gives them an advantage over obligate autotrophic protists in Lake Zurich that are depending on favorable light conditions and the availability of nutrients. In the cold season, mixotrophs can meet their energy demand by using bacteria as a primary energy source. In the warm season, the energy demand of mixotrophs can be covered by performing photosynthesis even under nutrient shortage. Mixotrophic chrysophytes might continue to feed on bacteria at low rates in the warm season to ensure that their energy demand is covered. However, statistically significant correlations between mixotrophic chrysophytes and bacteria were only observed in the cold season and not in the warm season network. Therefore, this scenario needs further testing in lab experiments.

### 4.4. Assessing Ecosystem Resilience of Lake Zurich with Protistan Community Networks

Our results showed that biotic and abiotic associations within protistan plankton communities in Lake Zurich are more complex and more robust towards disturbances in the cold compared to the warm season (Table 2, Figure 3 and Figure 4). Thereby, the climate-change-induced stress in Lake Zurich can be categorized as a press disturbance [15], since it has been continuously affecting the ecosystem for years (and will continue to do so). A study of vertebrate communities in terrestrial ecosystems illustrated how network analyses of complex co-occurrence patterns among species can assess the impact of climate change on ecosystems [27]. Based on their results, the authors developed a framework in which robustness and connectivity emerged as indicative network metrics for the susceptibility of an ecological community to climate change. Studies on bacteria [23,24,26,37], marine communities [42,43], and benthic macroinvertebrates [91] have successfully applied similar strategies for inferring the effect of environmental stressors from network analyses of community data. Different indicator measures exist for assessing the complexity of a co-occurrence network based on metabarcoding data [12]. Our results (Table 2) demonstrate that a larger input dataset does not necessarily lead to a network that comprises more nodes (here protists) and edges (here co-occurrences). Neither does a count of nodes or edges alone allow for concluding the complexity of a network. Such conclusions need to be drawn from the network topology, by putting the numbers of nodes and edges into context and inferring their distribution and connection patterns within the network. The density of a network is one such metric and displays how many of the potential edges (with regard to the total number of nodes) are effectively established. The higher the density, the more associations in the network, and the more complex and resilient to stressors the community network become [92,93]. Average path length, average degree, and diameter are indirectly linked to density and also evaluate the distribution of edges in a network. With more associations established among the same number of organisms in a network, mean path distance and diameter decrease, and the complexity of the network increases. Another indicator measure for the complexity of a network is modularity. If this measure is high, the network is partitioned into many groups in which protists are co-occurring with each other but less with nodes from other groups (similar to functional guilds) and the network complexity decreases. Although modularity was just marginally higher in the warm season protistan plankton community network of Lake Zurich, the lower level of complexity in the warm season can further be derived from the partitioning of its co-occurrence network into many more connected components (most of which consisting of only two nodes) than in the cold season co-occurrence network.

Since functional redundancy correlates with ecosystem resilience [94], we referred to network transitivity as an indicator measure for inferring functional redundancy in the protistan plankton community networks. This metric indicates redundant associations, by measuring the probability that two nodes are at the same time, directly and indirectly, connected [12]. The higher this probability, the more redundant are associations in the network and the more complex and robust towards the extinction of species becomes the protistan community. Finally, we also assessed ecosystem resilience by documenting the response of the network towards targeted attacks (Figure 4). The longer a community network can maintain its structure and buffer the loss of connectivity, the more robust and less vulnerable it is. Thereby, functional redundancy has a positive effect on network robustness, because of its buffering capacity on node removal from the network.

Ongoing climate change can accelerate the loss of ecosystem resilience by promoting changes in protistan plankton community networks. However, it is currently difficult to predict whether the changes will continue towards a collapse of the networks or whether they lead to another stable state with permanent changes in ecosystem function and services. The latter assumption finds support in reports about declining fish stocks and increased costs for drinking water purification [95,96]. Beyond large-scale consequences for ecosystems, co-occurrence networks obtained from large metabarcoding datasets proved beneficial for developing hypotheses about autecological species-specific relationships. Based on these hypotheses, future interdisciplinary approaches can be designed where limnology, ecology, and molecular as well as morphological taxonomic expertise are considered to elucidate specific key players and their function in aquatic microbial food webs.

## Figures and Tables

**Figure 1 microorganisms-09-00549-f001:**
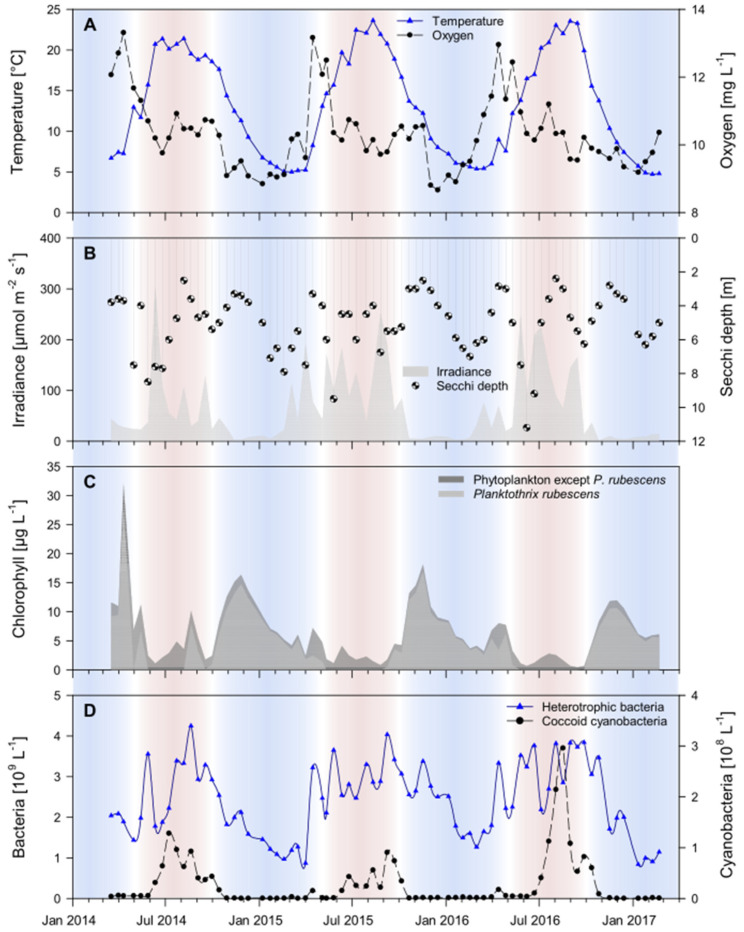
Time-series data of environmental parameters. Seasonally fluctuating parameters were measured throughout the three-year sampling campaign in the epilimnion of Lake Zurich. In all panels, cold season months are colored in blue and warm-season months in red. Panel (**A**) shows water temperature (blue line) and oxygen concentration (black line). Panel (**B**) shows irradiance (grey area) and the Secchi depth (data points). Panel (**C**) shows chlorophyll concentration of all phytoplankton (dark grey area) and the fraction of *Planktothrix rubescens*-related chlorophyll concentration among the total chlorophyll concentration (light grey area). Panel (**D**) shows cell counts of all heterotrophic bacteria (blue line) as well as of coccoid cyanobacteria (black line).

**Figure 2 microorganisms-09-00549-f002:**
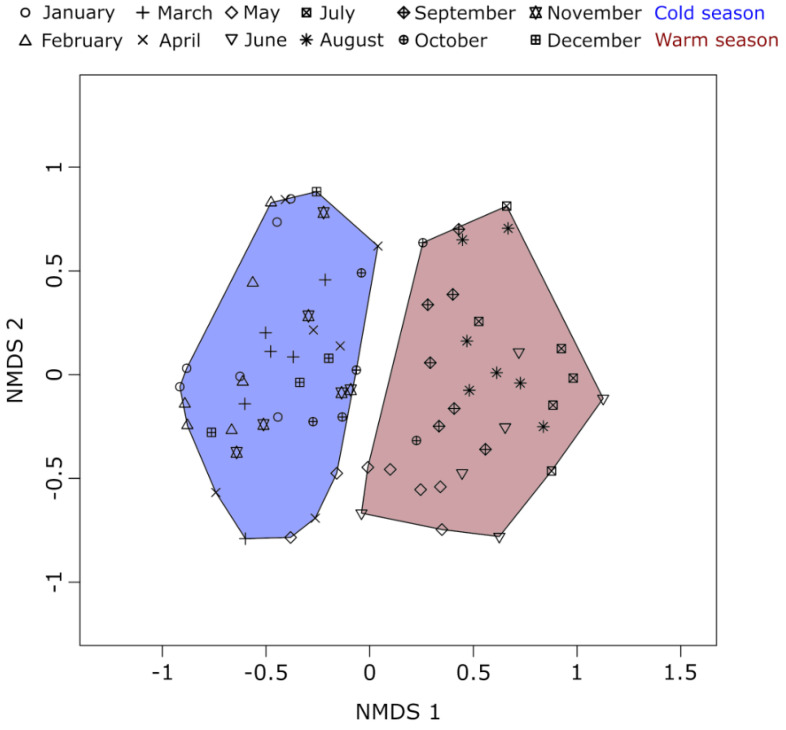
Non-metric multidimensional scaling of time-series data. The scaling is based on Bray–Curtis dissimilarity values of metabarcoding community data between each pair of samples. Each symbol represents a different month of sampling and each sample was classified into either cold season (**blue**) or warm-season (**red**). With few exceptions, samples from October to April were classified as cold season and samples from May to September as the warm season. The stress for the scaling was 0.1826.

**Figure 3 microorganisms-09-00549-f003:**
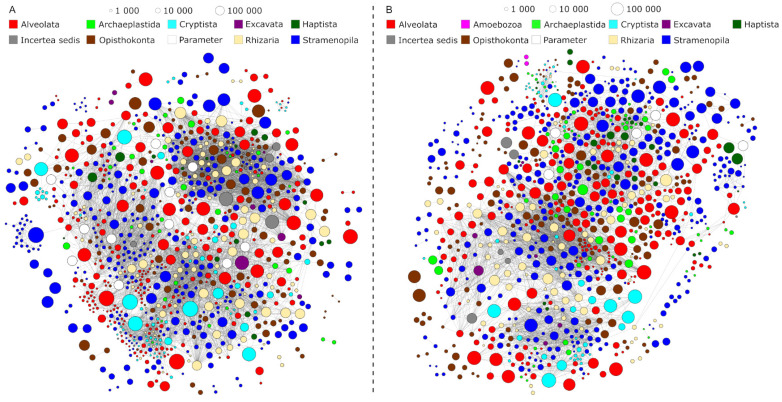
Seasonal protistan plankton co-occurrence networks. Panel (**A**) shows the cold season network, panel (**B**) the warm season network. Each node represents one network sequence cluster (NSC). Two nodes are connected by an edge if they were significantly co-occurring in the three-year time-series dataset of Lake Zurich. Node colors reflect the taxonomic assignment of NSCs on higher taxonomic levels. Node sizes reflect the read abundance of NSCs in the metabarcoding dataset.

**Figure 4 microorganisms-09-00549-f004:**
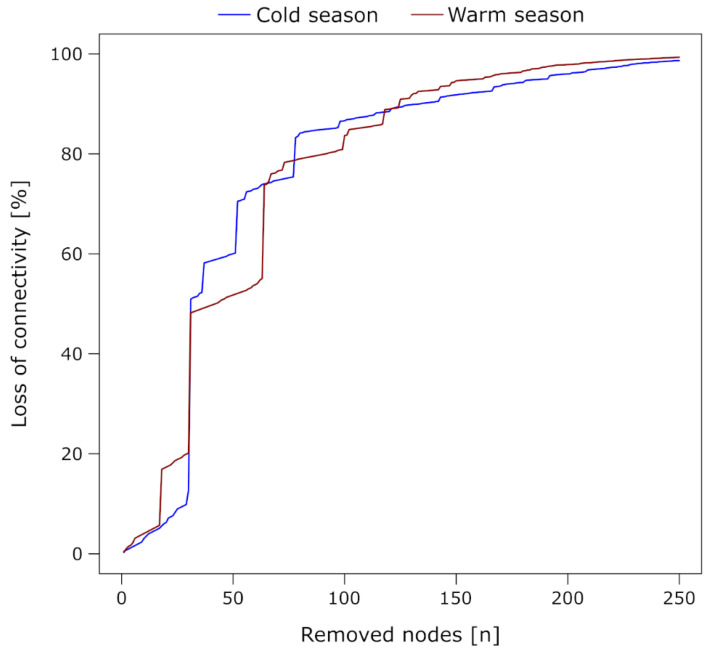
Robustness of the networks in both seasons towards a species extinction scenario. We stepwise removed the nodes with the highest betweenness centrality from the network and recorded the loss of connections (edges) in the network that was caused by this cascading attack. Displayed is the removal of the 250 nodes with the highest betweenness centrality from each network, after which the loss of connectivity was 98.67% in the cold season (blue) and 99.32% in the warm season (red). At this point, the networks had almost completely disintegrated and the removal of more nodes had only little effect.

**Figure 5 microorganisms-09-00549-f005:**
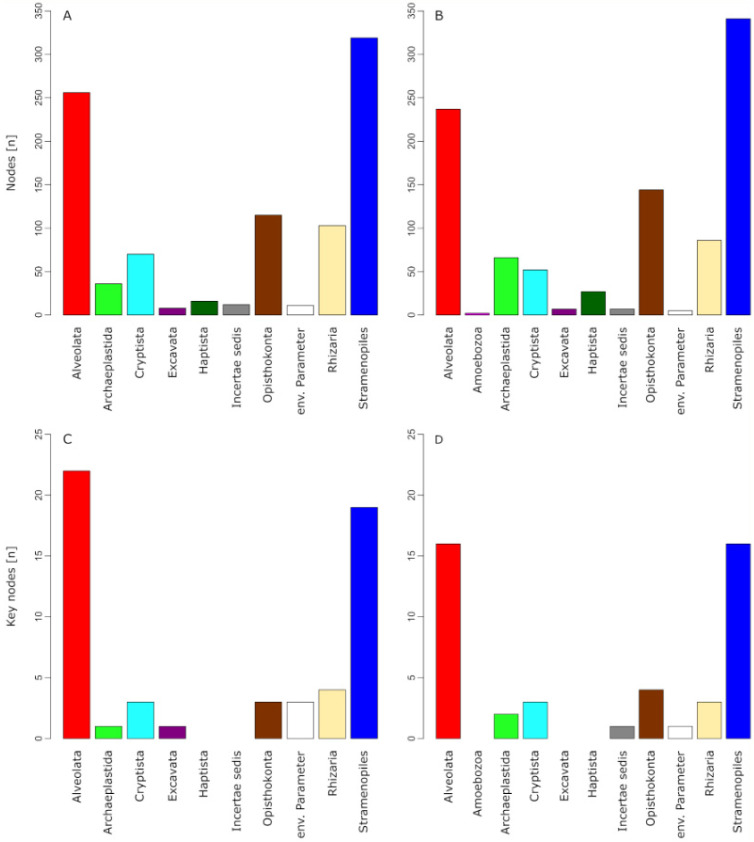
Community composition and key nodes of the co-occurrence networks in both seasons. Taxonomic assignment to higher taxonomic levels follows Adl et al. [62]. Panel (**A**) displays the community composition of the cold season network, panel (**B**) displays the community composition of the warm season network. Panel (**C**) displays the taxonomic affiliation of key nodes in the cold season network, panel (**D**) displays the taxonomic affiliation of key nodes in the warm season network.

**Figure 6 microorganisms-09-00549-f006:**
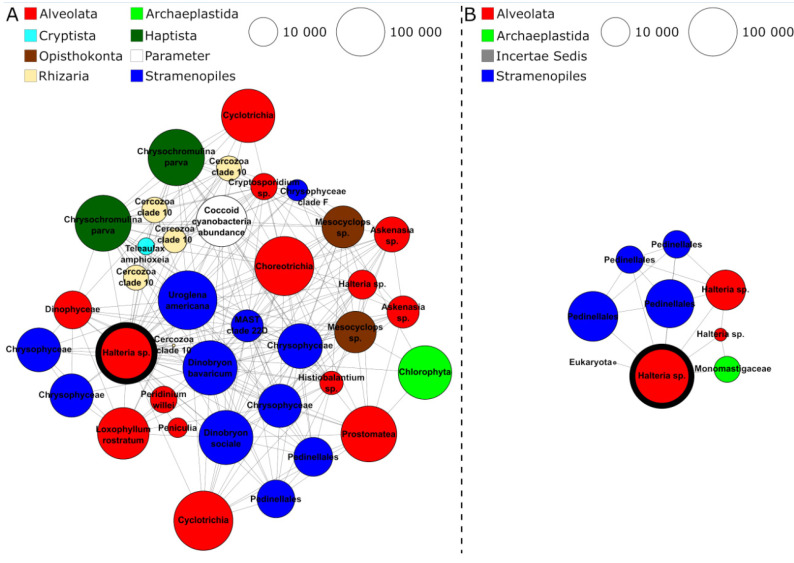
**Species-specific co-occurrence subnetworks of *Halteria* sp. in both seasons.** One network sequence cluster (NSC) assigned to the ciliate *Halteria* sp. was identified as a key node in the networks of both seasons. The *Halteria* sp. key node is highlighted by a bold black circle. All directly neighboring nodes of this *Halteria* sp. node are displayed in the subnetworks above. Panel (**A**) shows the subnetwork of the cold season, panel (**B**) the subnetwork of the warm season. Node colors reflect the taxonomic assignment of NSCs on higher taxonomic levels. Node sizes reflect the read abundance of NSCs in the metabarcoding dataset.

**Table 1 microorganisms-09-00549-t001:** Major limnological characteristics of Lake Zurich.

Parameter	Average (Minimum–Maximum)	Unit
Water temperature	12.7 (4.7–23.7)	°C
Air temperature	11.8 (−6.0–24.6)	°C
Secchi depth (water transparency)	5.0 (2.4–11.2)	m
Conductivity	266 (219–293)	µS cm^−1^
Oxygen concentration	10.6 (8.7–13.3)	mg O_2_ L^−1^
Oxygen saturation	100 (72–126)	%
Orthophosphate *	1.6 (0.0–3.2)	µg P L^−1^
Total phosphorus *	12.3 (7.0–25)	µg P L^−1^
Particulate phosphorus *	8.5 (3.0–21)	µg P L^−1^
Nitrate (NO_3_-N) *	434 (113–610)	µg N L^−1^
Ammonium (NH_4_-N) *	6.0 (2.3–22.6)	µg N L^−1^
Dissolved organic carbon (DOC) *	1.4 (1.1–1.7)	mg C L^−1^
Total chlorophyll *a*	6.6 (0.5–32.1)	µg Chl *a* L^−1^
Maximal depth	136	m
Total lake volume	3.3	km^3^
Total lake area	66.6	km^2^
Water retention time	1.2	years

**Table 2 microorganisms-09-00549-t002:** Key properties of the co-occurrence networks in both seasons.

	Cold Season Network	Warm Season Network
Input samples	38	35
Input NSCs	21,667	23,904
Spearman’s rho co-exclusion threshold	−0.59	−0.61
Spearman’s rho co-occurrence threshold	0.6	0.62
Edges (co-occurrences)	6872	5252
Nodes (NSCs)	924	963
Nodes of environmental parameters	11	5
Average degree	14.53	10.79
Average path length	4.64	5.46
Connected components (larger than 3 nodes)	41 [6]	68 [7]
Density	0.015	0.011
Diameter	12	17
Modularity	0.02	0.03
Transitivity	0.49	0.43

NSC: network sequence cluster.

## Data Availability

The data presented in this study are openly available at the Sequence Read Archive of the National Center for Biotechnology Information (NCBI) under project number PRJNA609412.

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
