# Peer review of "Lake Ecosystem Robustness and Resilience Inferred from a Climate-Stressed Protistan Plankton Network"

_microorganisms, 2021, doi:10.3390/microorganisms9030549_

Round 1

Reviewer 1 Report

The authors used a time series of three years collection combined with metabarcoding and ecological network analysis to describe the Zurich lake community, robustness, and resilience. Although they present a nice sampling and analysis they discuss nonexistent results. My major concern is about all climatic change discussions. In my opinion, all topic "Succession in the protistan plankton network of Lake Zurich is affected by climate change" may be removed. Three-year sampling did not show a variation that could be related to climatic change and the analysis of cascading attack scenario is also one speculation since we do not know if key species will disappear with climatic changes. Indeed, such species could be more abundant. And all these points may be considered in the speculative part of the discussion or, as suggested, can be removed.

Minor comments: as the file has no number lines I attached the file with my comments.

Author Response

The authors used a time series of three years collection combined with metabarcoding and ecological network analysis to describe the Zurich lake community, robustness, and resilience. Although they present a nice sampling and analysis they discuss nonexistent results. My major concern is about all climatic change discussions. In my opinion, all topic "Succession in the protistan plankton network of Lake Zurich is affected by climate change" may be removed. Three-year sampling did not show a variation that could be related to climatic change and the analysis of cascading attack scenario is also one speculation since we do not know if key species will disappear with climatic changes. Indeed, such species could be more abundant. And all these points may be considered in the speculative part of the discussion or, as suggested, can be removed. 
Authors: We appreciate the thoughts of reviewer 1 on this specific topic. Such thoughts are not uncommon as they are based on a widespread misconception on network theory, and above all, on how network results can be interpreted. The field of graph theory is still relatively new in microbial ecology and, due to its complexity, is not yet fully appreciated in our scientific community. The key point is to understand what networks are capable of and what is an overinterpretation. Therefore, we fully understand the reluctance of the reviewer to accept our conclusions, yet, we strongly disagree with the reviewer’s concerns. Following, we try to dispel these concerns. The reviewer observed correctly and with a sharp eye that there is hardly variation in the succession patterns which emerged from our dataset. However, this does not mean that climate change effects on the protistan plankton communities are not evident. On the contrary, at several occasions in the manuscript we discuss the evidence that the epilimnetic protistan communities we investigated in our study were continuously and severely affected by climate change. To mention one example, the community composition and succession patterns unveiled by our dataset diverged from traditionally observed community composition and succession patterns in Lake Zurich (Eckert et al. 2012 Environ Microbiol 14:794-806; Posch et al. 2012 Nat. Clim. Change 2:809-813; Yankova et al. 2016 Hydrobiologia 776 125-138; Yankova et al. 2017 Sci. Rep. 7:1-9). These traditional patterns have been documented for decades, but were only recently compromised due to lake warming, which is documented in several high quality publications and widely accepted by the scientific community of freshwater ecologists (Livingstone et al. 2003 Clim. Change 57, 205-225; Eckert et al. 2012; Posch et al. 2012; North et al. 2014 Glob. Change Biol. 20, 811-823; Yankova et al. 2016; Yankova et al. 2017). We do not only confirm this existing knowledge, but our new analyses extend the current state of knowledge by providing a hypothetical framework of how protistan plankton communities react to the drastic environmental changes and in how far they are able to buffer e.g., the decline of phytoplankton. And this is one of the strengths of network analyses, which allow the inference of hypotheses from extremely complex and massive data sets (time-series of millions of individual variables (OTUs in this case)). Because our manuscript has a research focus rather than a methodological focus (such as explaining and demonstrating the statistical power of network theory), it is unfortunately not possible to go more into methodological considerations without diluting the new main scientific findings that we convey to the scientific community. However, in the introduction section and also in the discussion section we at several occasions refer to very strong papers that have discussed the power of network theory in detail. We would like to use the opportunity and refer the reviewer once more to this body of literature, especially with context to network theory and climate change (e.g., Dunne et al. 2002 Ecol Lett. 5:558-567; Kéfi et al. 2016 PLOS Biology 14:e1002527; Karimi et al. 2017 Environ. Chem. Lett. 15:265-281; Mandakovic et al. 2018 Sci. Rep. 8:5875; de Vries et al. 2018 Nat. Commun. 9:3033). But we also agree with the reviewer that a similar dataset from before the onset of lake warming in Lake Zurich would be a perfect complement for our analyses (and this may be what the reviewer is looking for). Unfortunately, such a dataset is not available. Yet, even without such a dataset, the data we present do not lose their value and are still capable of telling a strong story (again, thanks to the power of network theory, please, see papers cited above). Furthermore, there are still other very valuable and long-term documentations of protistan plankton communities in Lake Zurich, to which we compared our results and from which the remarkable changes of recent years become more than obvious. Most of these previous studies are based on microscopical live observations; one study even created a co-occurrence network of heterotrophic and autotrophic protists from cell counts (Posch et al. 2015 Front. Microbiol. 6 1289:11289:14). We considered the results of all these studies while discussing our own data and are certain that our study provides unprecedented insights into the protistan plankton communities of an ecosystem which currently suffers from climate change. In the following, we present our detailed point-by-point response to the reviewer’s comments, which we hope the reviewer will find highly interesting and convincing. 
Minor comments: As the file has no number lines I attached the file with my comments.

Authors: We also realized that line numbers were missing from the edited manuscript file in the submission platform. To solve this issue and ease the review process, we included line numbers in the file provided online by Microorganisms and listed below all comments of reviewer #1 with line numbers referring to the initial manuscript version (if changes were made, line numbers after revision are given in parentheses).  
 Reviewer 1: l. 15: Potential consequences

Authors: While we agree that network analyses can in first place be used to develop hypotheses via statistically significant correlations, several studies have shown that these hypotheses correctly predicted the impact of environmental change on communities. This point is explicitly outlined in the discussion paragraph ‘Placing cooccurrence networks into perspective’: ‘Although co-occurrence networks analyses are limited to deriving potential species association patterns exclusively from significant correlations in species (or ASV, OTU) abundance matrices rather than from experimental observations, numerous studies have shown that such associations can be correctly predicted. More importantly, they allow for drawing valuable conclusions on community- or ecosystem-level [12,14,17–19,22–24,43].’ (l. 460-464), as well as: ‘Nevertheless, we point out that eventually, conclusions drawn from network analyses require rigorous testing in specifically designed experimental studies with e.g., cultivated species, or with targeted observations in natural systems. Network analyses provide, thus, the hypothesis framework to build explanatory models and fuel further research [19,73].’ (l. 475-479). Nevertheless, to address the reviewer’s suggestion, we rephrased to: ‘Network analyses of biological communities allow for identifying potential consequences of climate change on the resilience of ecosystems and their robustness to resist stressors’ (l. 15-17).  
Reviewer 1: l. 27: Are they really became?

Authors: We presume the reviewer’s question aims at clarifying if this conclusion is based on network analyses or on live observation of the organisms. Our microscopical observations were in first place conducted to infer cell counts of e.g., eukaryotic phytoplankton and not to infer interactions (because potential interactions were only known after the networks were created and not while sampling was still conducted). Cell counts clearly documented a decrease of typical phytoplankton as predicted, e.g., by the Plankton Ecology Group (PEG) model (Sommer et al., 1986 Arch. Hydrobiol. 106:433-471; Sommer et al., 2012 Annu. Rev. Ecol. Evol. Syst. 43:429-448). Logically, the predators had to turn to other prey organisms as indicated by our network analyses and thoroughly discussed in our manuscript (l. 558-578). However, we rephrased the sentence to clarify that our conclusion is drawn from the results of network analyses: ‘Instead of phytoplankton, bacteria and nanoflagellates were main prey organisms associated to key predators (ciliates), which contrasts traditional views of biological associations in lake plankton.’ (l. 26-28).   
Reviewer 1: l. 41: Typo corrected

Authors: Done, the corrected sentence reads: ‘A major challenge in revealing the complexity of these interactions is to account for temporal shift in protistan community structures.’ (l. 41-42). 
Reviewer 1: l. 46: It is possible identify interactions from a singular sampling of multiple sites. 
Authors: Regarding the reviewer’s comment, we first want to point out that our statement was clearly about ‘Singular samplings of individual sites’ (l. 45-48). Maybe this statement was misinterpreted by the reviewer towards ‘Singular samplings of multiple individual sites’, which was not our intention. Nevertheless, we would like to address the possibility mentioned by the reviewer, as well: Indeed, there are several studies which sample along spatial gradients (i.e., multiple sites) rather than conducting a time-series sampling strategy. By no means do we intent to discredit spatial sampling strategies for inferring potential interactions. But there is a current debate if spatial samplings, even when conducted at multiple sites, can accurately predict interactions. Please see Blanchet et al. (2020 Ecol Letters 23:1050-1063) for a thorough discussion of this topic (e.g., ‘In this paper, we present a series of arguments based on probability, sampling, food web and coexistence theories supporting that significant spatial associations between species (or lack thereof) is a poor proxy for ecological interactions’). In short, Blanchet and colleagues argue that sampling along spatial gradients only gives information about the presence and absence of organisms at particular sites, but gives no information about how abundances of organisms fluctuate over time. The latter is a much stronger information, since we know several models that explain how organism abundances correlate with each other (e.g., LotkaVolterra model) or with an environmental parameter over time. Therefore, we are convinced that our statement in this sentence is correct, especially since we explicitly refer to ‘habitat changes’: ‘Singular samplings of individual sites only provide a snapshot of a specific moment in time of the protistan community under study, and, therefore, do not allow to infer the complex interactions in these communities and their reaction to habitat changes.’ (l. 45-48). Singular samplings without a temporal dimension, no matter if conducted at an individual or at multiple sites, cannot be used to infer reactions of communities to habitat changes. Inferring such reactions demands time-series data that document the changes in the respective habitat.  
Reviewer 1: l. 58: Potential biological interactions

Authors: We adopted the reviewer’s suggestion. The revised sentence reads: ‘A statistical approach to infer potential biological associations from time-series data are co-occurrence network analyses [12,16–22].’ (l. 59-60). 

Reviewer 1: l. 212: Symbols are hard to follow, maybe include different colors... Authors: We had created several versions of this figure to determine the best visualization of the results for the readers. In a previous pre-submission version of the manuscript, we included 12 different colors, one for each month (please, see the figure following our response to this comment). These different colors, however, were no improvement compared to the current version of Figure 2 in which different symbols are used for the different months. Especially for people which have problems to distinguish different colors, the use of symbols is favorable compared to the use of colors. Putting colored data points into colored areas does not help with regard to this issue, or improve the figure, either. The most important result of the NMDS is the separation of cold season and warm season samples. We think that the current version of Figure 2 is the best presentation to infer this separation.   
Reviewer 1: l. 232: Connor et al. 2017 explain the null models strategy for this analysis. It is the recommended citation!

Authors: Yes, we fully agree with the reviewer that the null model strategy was developed by Connor et al. (2017; Plos One 12(5): e0176751). Therefore, we cite Connor et al. (2017) at multiple occasions throughout the manuscript, also for introducing the null model strategy in this respective paragraph of material and methods (l. 231). The specific citation the reviewer is referring to, however, is about the implementation of the null model strategy into the script NetworkNullHPC by Guillaume Lentendu. The reference Lentendu & Dunthorn (2020; bioxRiv; https://www.biorxiv.org/content/10.1101/2020.04.27.063685v1) contains a step-bystep explanation of this implementation, that our readers might find useful. We have no objections to include Connor et al. (2017), at this point again and revised the sentence to: ‘In addition, only nodes and edges which were persistently detected in the majority of permutations were retained for creating a consensus network (see [16] and [66] for detailed descriptions of the complete null model strategy).’ (l. 239-241). 

Reviewer 1: l. 282-283: Why did not you run a network with all data to see if there is two modules explained by the season?

Authors: In fact, we performed the analyses the reviewer is asking for in a very early stage of this study, but realized that it is not suited for addressing our hypotheses. There are methodological reasons why the reviewer’s expectations of two modules explained by season cannot be met. A network spanning all seasons will show correlations between pairs of protists that only occurred in one season and between pairs of protists which occurred throughout the year. Such a network, however, cannot identify changes in correlation patterns of a particular protist between the seasons. Even worse, since seasonally restricted correlations only span a fraction of the samples, they might not be considered significant by the null model strategy and will therefore not be included in the network. But ecological considerations tell us that an organism which can be detected throughout the year might depend on different abiotic and biotic associations in different seasons. Indeed, we can see such seasonally changing associations patterns in the subnetwork of Halteria sp. (see Figure 6 and l. 432-447 of the results, as well as l. 579-596 of the discussion). Splitting the dataset into seasonal subnetworks thus allowed for revealing season-specific correlations which could not have been detected by one network spanning all samples. At the same time, correlations between pairs of protists that only occurred in one season and between pairs of protists which occurred throughout the year will still be represented in the seasonal subnetwork. Ordination methods based on (beta-diversity) dissimilarity scores such as NMDS (Figure 2) are better suited for revealing differences in sets of samples. The results of our NMDS (Figure 2) showed a clear and statistically supported (stress: 0.1826; l. 226) separation of cold season and warm season samples. If such a separation would not have been detected, we would not have split the dataset; if an environmental parameter other than seasonal difference would have explained a separation of samples, we would have relied on this other parameter. Making use of this kind of previously acquired knowledge for network analyses is much more straightforward when constructing networks, compared to running an inclusive analysis and trying to identify patterns in retrospective. As an additional benefit, our strategy reduced the computational demand for network analyses, since this demand rises exponentially with the number of species and sites in the input dataset. 
Reviewer 1: l. 429: It is a justification for the analysis, and not the general results in perpectives. I think it should be the end of discussion and not the begin... Authors: This is more a matter of taste than a scientific decision and we do not necessarily agree regarding the position of this paragraph. Here is our rationale: The paragraph was specifically included for readers who are not familiar with network analyses or are sceptic about their value in ecological studies. Our experience in publishing studies about network analyses (e.g., Forster et al., 2015 BMC Biol 13, 1:11:16; Qu et al., 2021 Mol Ecol 30(4), 1053-1071) shows that incorporating such a statement is essential. As of late, there have been discussions about the application of co-occurrence network analyses and their interpretations (Blanchet et al., 2020; Carr et al., 2019 ISME J 13(11): 2647-2655). In the light of these discussions, we consider it important to underline that network analyses are in first place statistically significantly supported approximations of associations in natural communities. Their results should not be overinterpreted as evidence of interactions, but represent strong hypotheses on which further research can be based. Several comments of the reviewer, such as indicating co-occurrences as ‘potential’ biological associations or stating that results of the cascading attack scenario are speculation, confirm our perception. Understanding the methodological restrictions of co-occurrence network analyses (and the means we applied to overcome them) before starting with the discussion of the results will help readers to put the information into perspective while reading. Whereas considering the methodological restrictions after delving into our discussion is in our opinion anticlimactic. 

Reviewer 1: l. 450: I am not convinced of this. This is not a result from this study, it is a speculation. It is ok to speculate about what the results could mean, but as it is presented it appear that it is a conclusion and it is not. The three year sampling did not show a variation that could be related to climatic change and the analysis of cascading attack scenario is also one speculation since we do not know if key species will disappear with climatic changes. Indeed, such species could be more abundant. And all these point may be considered in the speculative part of the discussion.

Authors: We understand skepticism towards mis- or overinterpreting results, especially when it comes to such delicate topics as climate change and its impact on natural ecosystems. In this particular case, however, we can affirm that the data record speaks for itself. Please, see also our first response to the general comment made by this reviewer. In more detail to this specific comment: It is a matter of fact that lake warming has led to rising surface water temperatures, which have interrupted complete water turnover in Lake Zurich (documented by the environmental parameter profiles in our study and in multiple previous studies: Livingstone et al. 2003; Posch et al. 2012; North et al. 2014; Yankova et al. 2016; Yankova et al. 2017). To reinforce this argument, we have included more information on environmental parameter profiles in the revised manuscript. For instance, an additional table on limnological characteristics of Lake Zurich (new Table 1), an additional paragraph in the results in which the parameter profiles are presented (l. 287-302) and an additional paragraph in the discussion in which we expand our previous thoughts about those parameters which correlated with protists in the networks (l. 515-527).  It is also a matter of fact that the interruption of water turnover has led to an oligotrophication of the epilimnion in Lake Zurich, because nutrients accumulate in deeper water layers and are not transported back to the epilimnion. This finding is confirmed by our environmental parameter profiles and the multiple studies which had previously been conducted in Lake Zurich (Posch et al. 2012; North et al. 2014; Yankova et al. 2016; Yankova et al. 2017). We hope that our more extensive presentation of environmental data is beneficial for retracing this conclusion, as well.  It is a further matter of fact that the climate change-induced environmental parameter changes exhibit impact on (protistan) plankton communities in Lake Zurich. This impact is highlighted by the emergence of Planktothrix rubescens as the dominant autotrophic organism in the lake. Figure 1 of our study documents this dominance via chlorophyll measurements. P. rubescens-related chlorophyll measurements further correlated with several protists in the cold season, but not in the warm season network. More details on the climate change-induced emergence of P. rubescens can be found in Posch et al. (2012) and Yankova et al. (2016).   Finally, it is also a matter of fact that protistan phytoplankton spring blooms, which were investigated in Lake Zurich for decades and usually established between March and May (Eckert et al. 2012), have been largely absent during the recent period of lake warming (Posch et al. 2012; Yankova et al. 2017). Chlorophytes, chryptophytes and diatoms used to be the main phytoplankton organisms in Lake Zurich, acting as primary producers and serving as prey for heterotrophic protists. As evident from Figure 5 and discussed in lines 541-557, these taxa played only minor roles in our protistan plankton co-occurrence networks of both seasons. Furthermore, if stable phytoplankton spring blooms would have been established, these would have dominated the community profiles of the respective samples. Our NMDS analyses, however, did not show a third cluster of spring samples in addition to the cold season and warm season samples. Instead, the separation between cold season and warm season samples was determined between April and May, exactly when spring blooms had traditionally established. We realized that this last point had not yet been considered in the manuscript and added the following statement: ‘A third seasonal group, which comprised distinctive spring sample communities could not be observed.’ (l. 309-310).   The above outlined line of arguments is an integral part of our manuscript and clearly shows that climate change does indeed affect the ecosystem of Lake Zurich on multiple levels. And there is clear evidence that, unlike before climate change drastically altered the ecosystem, protistan phytoplankton succession patterns in the lake do no longer follow the predictions from the PEG model (Sommer et al. 1986; Sommer et al. 2012). Within this context, our study employs network analyses to i) detect patterns within protist plankton communities that differ from the traditionally expected succession in Lake Zurich, ii) investigate these differences and provide explanations for their occurrence under consideration of environmental parameters that are unambiguously linked to climate change and iii) develop hypotheses on further consequences for the protistan plankton communities; for instance, which organisms might occupy replace phytoplankton in the seasonal succession as primary producers and prey organisms for other protists. Concerning the species extinction scenario, the reviewer misinterpreted the underlying concept. This analysis does not aim at resolving the absolute impact of the loss of a specific species. Neither does it make any claims about how likely or realistic the loss of a specific species is. It is a purely hypothetical scenario which allows for inferring a network’s response as a whole to the stepwise removal of nodes. Attack scenarios which assess the robustness of a network are widely applied in network analyses. The method provides valuable information for ecologists by indicating the potential impact of disturbance on ecosystems, as documented by several benchmark publications (e.g., Albert et al. 2000 Nature 406:378-382; Dunne et al. 2002; Gao et al. 2016 Nature 530:307-312). Since both reviewers were unfamiliar with the analysis, we added a brief statement to clarify the concept of targeted attack scenarios: ‘The method is based on a hypothetical scenario to assess the large-scale response of a network as a whole, but does not take the likelihood of a specific species’ extinction into account or allows inferences about this likelihood.’ (l. 276-279). We are positive that this statement will lead to a better understanding of the scenario and thus, will help readers to put the results into perspective. 

Reviewer 1: l. 458-459: No, you showed a natural succession that keep similar in the three years! How climatic changes will impact these communities were not showed, it is speculative.

Authors: Please see our previous answers to this point. Thanks to several decades of lake ecology research in Lake Zurich, there exists a profound knowledge about the protistan plankton communities, about microscopically observed interactions within these communities, and annual succession patterns of protists. Moreover, the PEG model (Sommer et al. 1986, 2012) delivers widely accepted expectations and explanations of natural succession in lake ecosystems. While the reviewer is correct that the succession patterns detected in our study were similar throughout the three years, they were unambiguously different compared to historic data (including previous network analyses, see Posch et al. 2015, cited e.g., in l. 566) that were obtained before climate change had started to alter the ecosystem. All of our results (environmental parameter measurements, beta-diversity analyses and co-occurrence networks) corroborate these differences and explain in how far association patterns in protistan plankton communities are impacted by increasingly diminished roles of essential groups of organisms (phytoplankton) and of increasingly changing environmental parameters.  Reviewer 1: l. 473: Please report this result.  Authors: We appreciate this suggestion, since we are positive that it will be beneficial for the interpretation of our study and for further clarifying the impact of climate change on Lake Zurich. Therefore, we decided to include an additional table in the revised manuscripts (new Table 1) which covers measurements of phosphorous, nitrogen and carbon compounds. The same set of data can also be retrieved from the reference cited in this sentence, which was marginally edited (added Table 1 to the information in parentheses) and now reads: ‘Thermocline-induced incomplete water turnovers resulted in a nutrient depletion in the epilimnion of Lake Zurich during our investigation period (2014-2017; see Table 1 and Yankova et al. [45])’ (l. 503-505). 

Reviewer 1: l. 484-486: Your results showed a return of communities after each season. Reviewer 1: l. 486-488: It was not showed! Reviewer 1: l. 488-498: I did not see this "suggestion". 
Authors: Because these three comments relate to the same paragraph and content in our manuscript, we decided to address them within the same response:  Please be careful here. Our statement in this commented paragraph is explicitly about phytoplankton communities, not about the entire protistan plankton community: ‘Throughout the time of lake warming in Lake Zurich, phytoplankton communities have not been able to recover or to adapt to the new environmental conditions, as evident from the continuous non-initiation of spring blooms. Thus, the phytoplankton community is neither able to resist the ecosystem disturbance induced by climate change nor to re-cover to a stable state during the three-year observation cycle of this study. This suggests that both ecosystem robustness and resilience in Lake Zurich are compromised.’ (l. 528-533). Regarding the results of our study and previous reports from Lake Zurich (e.g., Posch et al. 2012; Yankova et al. 2017) there exists ample evidence that phytoplankton communities continually fail to establish spring bloom communities in Lake Zurich due to the lake warming effects. Such communities had been successfully established, however, before climate change started to dramatically alter Lake Zurich. The diminished role of phytoplankton becomes evident from our parameter measurements (Figure 1, please compare the total chlorophyll measurements against the chlorophyll measurements of Planktothrix rubescens), the absence of spring bloom communities (spring samples did not cluster together in Figure 2) and the taxonomic composition of the network communities (Figure 5, the networks comprised very few chlorophytes, cryptophytes and diatoms, which had formed phytoplankton blooms before climate change effects became to dramatic in Lake Zurich). All of this information is outlined in the results (l. 292-297, l. 305-309, l. 393-412) and the discussion (l. 481-514, l. 541-578, l. 597-608) of our manuscript.  
Reviewer 1: l. 541: Potential role. Authors: There exists a wealth of literature which shows that ciliates indeed link different trophic level in lake ecosystems (e.g., Sherr & Sherr 1987 Nature 325:710711; Sherr et al. 1988 Hydrobiologia 159:19-26; Müller 1989 Microb. Ecol. 18:261-273; Pernthaler 2005 Nat. Re. Microbiol. 3:537-546). The results of our network analyses find full support by this concept. Although we are perfectly convinced by our results, we decided to comply the reviewer’s request. The revised sentence reads: ‘This result illustrates the potential role of ciliates in linking energy fluxes between different trophic levels [1].’ (l. 585-586). 

Reviewer 1: l. 571-571: It is not, the difference is the season, develop better this sentence.

Authors: Reviewer 2 asked for a revision of the same sentence (because of different reasons, though). We thus rephrased to: ‘A study of vertebrate communities in terrestrial ecosystems illustrated how network analyses of complex co-occurrence patterns among species can assess the impact of climate change on ecosystems [27]. Based on their results, the authors developed a framework in which robustness and connectivity emerged as indicative network metrics for the susceptibility of an ecological community to climate change.’ (l. 614-619). We are positive that this revision of the sentence also addresses the comment of reviewer 1.  
Reviewer 1: l. 574-576: Lost sentence, remove.

Authors: The sentence originally referred to another sentence which was rephrased according to the previous comment (l. 614-619). Because the sentence does not fit anymore to the revised version of the manuscript, we followed the reviewer’s suggestion and removed it from the manuscript. 

Reviewer 1: l. 576-578: Your results are very similar, make this afirmation impossible.

Authors: With all due respect to the reviewer’s opinion, we beg to differ. The sentence reads: ‘Our results (Table 2) demonstrate that a larger input dataset does not necessary lead to a network that comprises more nodes (here protists) and edges (here co-occurrences).’ (l. 625-627). There are 1,620 more edges in the cold season (n=6,872) than in the warm season (n=5,252) network (Table 2). These are notably different numbers, which can hardly be considered ‘similar’. 

Reviewer 1: l. 612-613: References?

Authors: We thank the reviewer for pointing out that references were missing for this statement. In the revised manuscript version, we included Jeppesen et al. (2012 Hydrobiologia 694:1-39) and Grafton et al. (2013 Nat. Clim. Change 3(4):315-321) to solve this shortcoming. Accordingly, the revised sentence reads: ‘The latter assumption finds support in reports about declining fish stock and increased costs for drinking water purification [95, 96].’ (l. 662-663).  

Reviewer 2 Report

This manuscript has a completed form analyzed and discussed using the protistan plankton network to evaluate lake ecosystem robustness and resilience. The manuscript is quite well written, and in the introduction, the discussion of the research results as well as the background and purpose of the study is described in conjunction with the existing literature. I thank the author for reading this interesting topic and manuscript well. Questions and questions raised while reading this manuscript are as follows.

Major comment

  1. Although the authors said they investigated and analyzed the protistan plankton network representing the lake ecosystem, the plankton was collected from only epilimnion. Can Epilimnion represent Lake Zurich's plankton network?
  2. The discussion mainly refers to plankton community networks, and the explanation of their impact or robustness on various environmental variables were understated. As mentioned in the Introduction section, time-series data in plankton community are complex outputs for environmental variables and interactions.

Minor comment

  1. Introduction

I have no question in this part. The introduction section is well written, and the background and purpose are well explained.

  1. Material and Methods

- Why was plankton commnuty sampling only investigated in epilimnion? In lakes, plankton commnuty are distributed in various spaces (vertically or horizontally) as well as epilimnion.

- Add the table on limnological characteristics (climate, hydrological characteristics, and water quality, etc.) for Lake Zurich. The information can increase readers' understanding of this manuscript.

- All samples were processed as previously described in Qu et al. [50]. I recommend explaining this sampling method in more detail. It can reduce the discomfort of readers interested in manuscripts.

- Please provide the range of plankton community in the manuscript suggests. The removal of large zooplankton using 150 µm plankton nets may still include smaller rotifers, some cladocerans (Bosmina) and nauplii. The authors re-filtered it back into the 0.65 µm membrane filter.

- The results of the environmental parameters and NMDS analysis should be presented in the 'Results' section. In materials and methods, the author explains that these two factors were directly measured or analyzed.

  1. Discussion

I suggest explaining this sentence in more detail for a better understanding.; Following a framework proposed from network analyses of vertebrates in terrestrial ecosystems [27],……..

Author Response

This manuscript has a completed form analyzed and discussed using the protistan plankton network to evaluate lake ecosystem robustness and resilience. The manuscript is quite well written, and in the introduction, the discussion of the research results as well as the background and purpose of the study is described in conjunction with the existing literature. I thank the author for reading this interesting topic and manuscript well. Questions and questions raised while reading this manuscript are as follows. 
Authors: We thank the reviewer for these kind words and are very pleased about the appreciation of our work. Please find our point-by-point answers to all questions and comments below 
Major comment 
1. Although the authors said they investigated and analyzed the protistan plankton network representing the lake ecosystem, the plankton was collected from only epilimnion. Can Epilimnion represent Lake Zurich's plankton network? 
Authors: The reviewer raises a valid question. As mentioned in the manuscript (l. 92), our sampling strategy was restricted to the epilimnion of Lake Zurich. While our findings are therefore restricted to epilimnetic plankton communities, they still lead to consequences for the whole lake ecosystem. Because plankton biomass and diversity are highest in the epilimnion and key ecosystem functions, such as primary production, are mainly performed in the epilimnion (e.g., Gaedke 1992; Limnol Oceanogr 37(6): 1202-1220), we expected the most severe effects of environmental change on protist communities in this layer of Lake Zurich’s water body. Environmental parameters affected by climate change, such as increased water temperature and a decrease in nutrients, are most severe in the lake epilimnion (e.g., Yankova et al. 2017). Nevertheless, the reviewer is correct to ask for a more precise statement in the manuscript. We decided to clarify the restriction to the eplimnion at several occassions (e.g., l. 106, l. 306, l. 376) and added the following statement to the introduction: ‘We inferred the protistan plankton co-occurrence network of the epilimnion in Lake Zurich 
(hereafter referred to as the protistan plankton community network) from a three-year time-series DNA metabarcoding dataset with biweekly sampling.’ (l. 92-94). This clarifies our restriction to the epilimnion; at the same time, we avoid multiple repetitions of the expression ‘epilimnetic protistan plankton co-occurrence network’, which, in our opinion, would not improve the readability of the manuscript. 
2. The discussion mainly refers to plankton community networks, and the explanation of their impact or robustness on various environmental variables were understated. As mentioned in the Introduction section, time-series data in plankton community are complex outputs for environmental variables and interactions. 
 Authors: We greatly appreciate this comment, as it gave us the opportunity to elaborate more on the importance of environmental variables and especially on the different effects of specific variables in the cold season and the warm season network. Actually, we had included such a paragraph in an earlier pre-submission version of the manuscript, but then decided to eliminate this part to keep the manuscript at a reasonable length. Encouraged by the reviewer’s comment, such a paragraph is now included again in the revised version. This new paragraph is indeed important, as it also address a major comment of reviewer 1: it shows the impact of lake warmingtriggered environmental effects on biotic communities in the epilimnion of Lake Zurich and confirms previous observations reported in some of the cited publications. This new paragraph discusses in specific environmental parameters that are of high relevance in the context of climate change driven effects on deep temperate lakes: temperature, oxygen and nutrients (conductivity as a sum parameter). The new paragraph reads as follows: ‘The identification of water temperature as a major determinant associated with numerous protistan taxa especially in the cold season, underlines its influence on co-occurrence patterns among protists in the epilimnion. Also, the effects of other environmental parameters triggered by lake warming became obvious as they significantly correlated with protistan co-occurrence patterns. These parameters include for example oxygen (both saturation and concentration), which was linked to a different set of organisms in either season, which suggests that this parameter is one of the most important environmental variables associated with planktonic organisms in the epilimnion of lake Zurich. Even though the epilimnion in Lake Zurich was saturated with oxygen in the warm season, a simultaneous decrease in conductivity indicated fewer solutes and nutrients in the epilimnion during this time of the year. Consequently, fewer protists correlated with conductivity measures in the warm than in the cold season, when more solutes were available.’ (l. 515-527). Yet, we note that it is impossible to determine the robustness of a single parameter. Robustness is a concept which indicates the reaction of a network as a whole. To clarify this, we added the following statement to the manuscript: ‘The method is based on a hypothetical scenario to assess the large-scale response of a network as a whole, but does not take the likelihood of a specific species’ extinction into account or allows inferences about this likelihood.’ (l. 276-279). The closest approximation to what the reviewer is asking for, is to investigate the number of connections in which an  environmental parameter is involved (and which are lost when the node is removed). This number is equal to the degree of an environmental parameter node, which we report in Table S2. As outlined above, water temperature was the environmental parameter node with highest degree, which explains our focus on this parameter in the discussion. The effect of water temperature on the protistan planktonic communities and their succession in Lake Zurich is therefore indisputable, as also documented by previous non-network studies. 
Minor comment 
1. Introduction 
I have no question in this part. The introduction section is well written, and the background and purpose are well explained. 
Authors: We are very pleased about the reviewer’s positive feedback on our introduction.  
2. Material and Methods  
- Why was plankton commnuty sampling only investigated in epilimnion? In lakes, plankton commnuty are distributed in various spaces (vertically or horizontally) as well as epilimnion. 
Authors: Please see our answer above to major comment #1. We focused on the epilimnion because previous studies have shown that Lake Zurich’s epilimnion is most severely affected by climate change (e.g., Yankova et al. 2017) and because protistan plankton diversity and biomass (e.g., Gaedke 1992), as well as the resulting impact of protists on ecosystem functions (such as primary production) is highest in this layer of the lake. In the revised manuscript, we now explicitly state that our results are only inferred from epilimnion communities (e.g., l. 106, l. 306, l. 376). The reviewer is correct, though, that it would have been intriguing to cover several spatial distributions and compare the resulting networks. There are few scientists who would not be happy to extend their sampling strategies. However, our year sampling campaign was already extensive (72 samples in total) and we are unaware of a comparable timeseries dataset of protistan plankton communities in lake ecosystems. An even more extensive sampling could not have been conducted given our financial framework and manpower.  
- Add the table on limnological characteristics (climate, hydrological characteristics, and water quality, etc.) for Lake Zurich. The information can increase readers' understanding of this manuscript. 
Authors: We followed the reviewer’s suggestion and inserted an additional table (new Table 1, l.132) in the material and methods section, which summarizes the limnological characteristics of Lake Zurich. 
- ‘All samples were processed as previously described in Qu et al. [50].’ I recommend explaining this sampling method in more detail. It can reduce the discomfort of readers interested in manuscripts. 
Authors: There is a misunderstanding here. In the initial manuscript version, we had already provided all sampling details which are also described in Qu et al. 2021 (Mol Ecol doi:10.1111/mec.15776) (see l. 105-131). Although we rephrased the description of sample processing with regard to this reference, it contains essentially the same information; therefore, we included the reference Qu et al. 2021 to avoid potential critiques of self-plagiarism. Interested readers will find all information to repeat the sample processing of our study within our material and methods section. 
- Please provide the range of plankton community in the manuscript suggests. The removal of large zooplankton using 150 µm plankton nets may still include smaller rotifers, some cladocerans (Bosmina) and nauplii. The authors re-filtered it back into the 0.65 µm membrane filter. 
Authors: We are aware that our sampling strategy did not exclude smaller zooplankton organisms. This happened on purpose, since several zooplankton organisms are important predators of protists and, as such, they may add important information to the protistan plankton community network. As discussed in our manuscript, protists transfer energy between different trophic levels (e.g., preying upon bacteria while being prey of small rotifers or crustaceans, themselves). Network analyses help to indicate such trophic links, as outlined in the discussion of the ciliate Halteria sp., which was associated with both bacteria and the copepod Mesocyclops (l. 583-596). According to the reviewer’s suggestion, we added one more sentence about the since range of organisms which we analyzed in our study: ‘Filters containing planktonic organisms in the targeted size range of 0.65-150 µm were immediately transferred into a cryovial containing 1.5 ml RNALater (Qiagen, Hilden, Germany), placed in a refrigerator overnight and stored at -80°C until further processing.’ (l. 113-115). The information about the size range was already in the initial manuscript version and can still be found in the sentence leading up to this newly inserted statement (l. 110-112). 
- The results of the environmental parameters and NMDS analysis should be presented in the 'Results' section. In materials and methods, the author explains that these two factors were directly measured or analyzed. 
Authors: We have no objection to present both figures in the results section. To address this issue, we decided to first mention both figures in the results section of the revised manuscript (and consequently removed their first mentioning in material and methods). Concerning the environmental parameter profiles, we have added a short paragraph at the beginning of the results: ‘Throughout the three-year sampling campaign of our study, the environmental parameters in the epilimnion of Lake Zurich displayed re-occurring seasonal patterns (Figure 1, Table S1). These patterns allowed for distinguishing one season characterized by cold water temperature lasting from October to April, and one season characterized by warm water temperatures lasting from May to September. Although irradiance by sunlight was naturally higher in the warm season, chlorophyll concentrations and secchi depths were lowest, which is a clear indication that phytoplankton could not establish stable populations. Planktothrix rubescens-related phycoerythrin contributed only a small portion to the total chlorophyll concentration, while heterotrophic bacteria and coccoid cyanobacteria thrived during warm season conditions. By contrast, irradiance was low in the cold season, but total chlorophyll concentrations more than doubled because of massive increases of phycoerythrin-containing P. rubescens. These increases also led to less transparency of epilimnetic waters as documented by smaller secchi depths. Unlike P. rubescens, heterotrophic bacteria and coccoid cyanobacteria reached their respective minima in the cold season.’ (l. 287-302). Concerning the NMDS, we understand the reviewer’s opinion to present these analyses in the results section. However, the outcome of the NMDS is the main argument for splitting the sequencing data into a cold season and a warm season dataset and, thus, for creating two independent networks. Without this crucial information, readers would not be able to follow the logical flow of our analyses and our decision to split the dataset into one cold season and one warm season dataset would seem arbitrary and unjustified. Nevertheless, the outcome of the NMDS had already been outlined once more in the initial version of the results to clearly explain our line of argument to the reader (l. 305-312).  
4. Discussion 
I suggest explaining this sentence in more detail for a better understanding.; Following a framework proposed from network analyses of vertebrates in terrestrial ecosystems [27],…….. 
Authors: We rephrased the sentence according to the reviewer’s suggestion. The revised version reads: ‘A study of vertebrate communities in terrestrial ecosystems illustrated how network analyses of complex co-occurrence patterns among species can assess the impact of climate change on ecosystems [27]. Based on their results, the authors developed a framework in which robustness and connectivity emerged as indicative network metrics for the susceptibility of an ecological community to climate change.’ (l. 614-619). 

Round 2

Reviewer 1 Report

I am satisfied with the authors' actions or responses to my previous review.